# *SPAST* Intragenic CNVs Lead to Hereditary Spastic Paraplegia via a Haploinsufficiency Mechanism

**DOI:** 10.3390/ijms25095008

**Published:** 2024-05-03

**Authors:** Ewelina Elert-Dobkowska, Iwona Stepniak, Wiktoria Radziwonik-Fraczyk, Amir Jahic, Christian Beetz, Anna Sulek

**Affiliations:** 1Department of Genetics, Institute of Psychiatry and Neurology, 02-957 Warsaw, Poland; eelert@ipin.edu.pl (E.E.-D.); neurogenetyka@protonmail.com (I.S.); wradziwonik@ipin.edu.pl (W.R.-F.); 2Institute of Diagnostic Laboratory Medicine, Clinical Chemistry and Pathobiochemistry, Charité–Universitätsmedizin, 10117 Berlin, Germany; amir.jahic@charite.de; 3Department of Chemistry and Laboratory Medicine, Jena University Hospital, 07747 Jena, Germany; christian.beetz@centogene.com; 4Centogene, 18055 Rostock, Germany; 5Faculty of Medicine, Lazarski University, 02-662 Warsaw, Poland

**Keywords:** hereditary spastic paraplegia, microrearrangements, *SPAST* deletions

## Abstract

The most common form of hereditary spastic paraplegia (HSP), SPG4 is caused by single nucleotide variants and microrearrangements in the *SPAST* gene. The high percentage of multi-exonic deletions or duplications observed in SPG4 patients is predisposed by the presence of a high frequency of *Alu* sequences in the gene sequence. In the present study, we analyzed DNA and RNA samples collected from patients with different microrearrangements in *SPAST* to map gene breakpoints and evaluate the mutation mechanism. The study group consisted of 69 individuals, including 50 SPG4 patients and 19 healthy relatives from 18 families. Affected family members from 17 families carried varying ranges of microrearrangements in the *SPAST* gene, while one individual had a single nucleotide variant in the 5′UTR of *SPAST*. To detect the breakpoints of the *SPAST* gene, long-range PCR followed by sequencing was performed. The breakpoint sequence was detected for five different intragenic *SPAST* deletions and one duplication, revealing *Alu*-mediated microhomology at breakpoint junctions resulting from non-allelic homologous recombination in these patients. Furthermore, *SPAST* gene expression analysis was performed using patient RNA samples extracted from whole blood. Quantitative real-time PCR tests performed in 14 patients suggest no expression of transcripts with microrearrangements in 5 of them. The obtained data indicate that nonsense-mediated decay degradation is not the only mechanism of hereditary spastic paraplegia in patients with *SPAST* microrearrangements.

## 1. Introduction

Mutations in the *SPAST* gene fulfill a crucial role in hereditary spastic paraplegia (HSP) due to being found in almost 40% of inherited HSP patients and approximately 20% of sporadic patients (single individuals with the disorder in the family) [1,2]. More than 700 different pathogenic variants in the *SPAST* gene include missense, truncating, and multi-exonic deletions, rarely duplications. Intragenic copy number variants (CNVs) deleting or duplicating diverse combinations of exons in the *SPAST* gene comprise approximately 30% of all alterations. It has been shown that the *SPAST* gene sequence is enriched in *Alu* family members, which predisposes it to distinct disease-associated deletions [3,4,5].

An *Alu* element (or simply, “*Alu*”) is the most abundant transposable element in the human genome. Transposable elements are rare sequences of DNA that can move (or transpose) themselves to new positions within the genome of a single cell. A typical *Alu* element is an approximately 300 bp long transposable nucleotide sequence [6,7] found throughout the human genome [8]. The number of full-length or partial *Alu* elements in the human genome is estimated at over 1 million [9,10].

Among 56 previously described microrearrangements in the *SPAST* gene, 40 were mediated by an Alu-based mechanism, in some cases including a more complex mechanism [3,4,5]. The present study reports additional intragenic CNVs within *SPAST* together with a characterization of its architecture and transcriptional-level analysis. Our data support the non-allelic homologous recombination mediated by the *Alu* sequence within the *SPAST* gene as a main mechanism leading to the exonic deletions in this gene.

The protein encoded by the *SPAST* gene is a distinct microtubule-severing protein. The loss of spastin function due to the *SPAST* mutation has been the most frequent explanation for the flaws in cytoskeletal organization and microtubule transport damage resulting in the neurodegeneration of the corticospinal tracts observed in HSP patients [11]. In comparison, it has also been shown that one of the spastin isoforms presented in the nervous system may lead to axonal degeneration through the gain of function mechanism [12]. Our results, obtained from the quantitative analyses performed in the patients’ RNA isolated from the whole blood samples, revealed that the loss of function mechanism is not the only cause of SPG4 pathogenicity caused by exonic deletion of the *SPAST* gene, regardless of the gene part.

## 2. Materials and Methods

The study group comprises 69 individuals including 50 patients with hereditary spastic paraplegia, in whom the copy number variants within the *SPAST* gene have been detected in the previous study [13], and 19 healthy relatives. The studied patients originated from 18 unrelated families. The affected family members from 17 families were carrying different ranges of microrearrangements in the *SPAST* gene, whereas one individual has a single nucleotide variant in the 5′UTR of the *SPAST* gene. The DNA samples from unrelated individuals with unique CNV were used for breakpoint mapping and cDNA sequencing. Moreover, the *SPAST* gene expression analyses were performed using the RNA samples collected from 19 individuals (16 affected and 4 healthy relatives).

Details concerning families with microrearrangements in the *SPAST* gene are shown in Table 1 (Table 1).

Informed consent was obtained from each patient before the material sampling and genetic testing.

### 2.1. DNA

DNA samples from 69 individuals were collected. The DNA was extracted from the leucocytes using the MagnaPure (Roche, Basel, Switzerland) automated system according to the manufacturer’s instructions.

### 2.2. RNA

To obtain the RNA, whole blood samples were collected from 19 individuals using PAXgene Blood RNA Tubes (Qiagen, Hilden, Germany). The mRNA was extracted by the PAXgene Blood RNA Kit (Qiagen) according to the manufacturer’s recommendations.

### 2.3. Copy Number Screening

Large deletion and duplication in the *SPAST* gene were tested by the SALSA MLPA Probemix P-165 Assay (MRC Holland, Amsterdam, The Netherlands) using 100 ng of the genomic DNA samples. The MLPA reaction, including (1) DNA denaturation, (2) probe hybridization, (3) ligation, and (4) amplification, was performed using a Veriti Thermal Cycler (Thermo Fisher Scientific, Waltham, MA, USA). The MLPA reaction products were separated on the ABI 3130 (Thermo Fisher Scientific) using highly deionized formamide and a Gene Scan 500 ROX size standard (Thermo Fisher Scientific). The relative peak ratio was calculated using Coffalyser.Net Software v. 210604.1451 (MRC Holland, Amsterdam, The Netherlands).

### 2.4. Breakpoint Analysis by Long-Range PCR and Sequencing

Long-range PCR was performed to detect and sequence *SPAST* gene breakpoints. To select the deletion-specific products, as narrowly as possible, sets of primers complementary to the intronic sequence that presumably contains breakpoints were designed and long-range PCR was performed. Candidate PCR products were extracted from the gel, purified, and sequenced in both directions using the BigDye Terminator Kit (Thermo Fisher Scientific).

### 2.5. cDNA Sequencing

The mRNA isolated from blood samples collected in PAXgene Blood RNA Tubes was used for reverse transcription using the Transcriptor First Strand cDNA Synthesis Kit (Roche). The cDNA samples were then used for polymerase chain reaction and Sanger sequencing. For PCR reaction primers complementary to the *SPAST* gene, exonic sequences covering specific rearrangements were used. PCR products were cleansed by the QIAquck PCR Purification Kit (Qiagen) and sequenced using the BigDye Terminator Kit (Thermo Fisher Scientific). The sequencing products were detected by ABI 3130 capillary electrophoresis (Thermo Fisher Scientific).

### 2.6. Real-Time PCR

The mRNA samples isolated from the leucocytes were reversely transcribed using the Transcriptor First Strand cDNA Synthesis Kit (Roche). The cDNA samples were used for the quantitative real-time PCR reaction by LighCycler 480 Real-Time PCR System (Roche). Light Cycler 480 SYBR Green I Master was used for amplification. Primers complementary to the cDNA sequence of the *SPAST* gene designed for exons 1, 3, 6, 8, 12, and 15 enabled the amplification of three different products (Figure 1a). The actin gene *ACTB* was used as a reference one in the rt-PCR analysis. Relative quantitative analysis was performed to calculate the ratio of three different PCR amplification products in each individual carrying the *SPAST* gene deletion. Each reaction was performed in triplicate and the mean Ct was used for the relative quantification.

## 3. Results

### 3.1. SPAST Gene Deletion/Duplication

In 50 patients originating from 18 families, we detected deletion or duplication embracing either single or multiple exons of the *SPAST* gene by MLPA analysis. Twelve different deletions and one duplication were identified (Table 1).

### 3.2. Breakpoint Analysis and cDNA Sequencing

In patients with *SPAST* CNVs, the long-range PCR was performed to localize the exact range of rearrangement. The breakpoint sequence was detected for five different intragenic *SPAST* deletions, and one duplication, where the long-range PCR resulted in mapping a breakpoint junction. Unfortunately, applying the long-range PCR in patients carrying the deletion of the first or final exon of *SPAST* was limited due to the very broad, undetermined range of the deletion region—upstream or downstream of the gene. The breakpoint regions mapped for six different *SPAST* CNVs were sequenced. In all cases, the sequencing revealed an *Alu*-mediated microhomology at breakpoint junctions and flanking regions. The performed sequence analyses confirmed the non-allelic homologous recombination—NAHR as a mechanism of the microrearrangements (Figure 1b). Moreover, in the case of five *SPAST* gene exonic deletions and one duplication, cDNA was sequenced, and altered transcripts were detected (Figure 1c).

### 3.3. SPAST Gene Expression Analysis

The *SPAST* gene quantitative real-time PCR was performed in 14 SPG4 patients, and 4 healthy relatives, using three different pairs of primers for the *SPAST* gene in each individual. The *SPAST* gene expression analysis was performed to study if there are any changes in the *SPAST* gene expression level dependent on the part of the gene that was deleted. *ACTB* was used as a reference housekeeping gene for normalization. We assessed the relative quantification ratio by calculating mRNA levels in patients with *SPAST* gene microrearrangements in reference to healthy individuals. The primer pairs for the rt-PCR reaction were designed so that at least one pair was complementary to the part of the *SPAST* gene within the deletion. We assumed that this pair would only amplify the non-mutated allele of the gene. The quantitative ratio between different rt-PCR amplicons suggests a similar level of three different *SPAST* gene amplicons in five patients. However, individuals carrying (i) deletion of 4–7 exons; (ii) exons 5–7; (iii) exon 8; (iv) exons 8–17; (v) exons 10–17; and also (vi) exons 14–16 showed only a reduced ratio of amplicons complementary to the deleted exons (Figure 1d).

The obtained results of the relative gene expression analysis show that loss of function is not a mechanism occurring in all SPG4 patients with an intragenic deletion within the *SPAST* gene.

## 4. Discussion

Spastin protein, encoded by the *SPAST* gene, is a microtubule-severing protein. The severing of microtubules is important for the ongoing transport as only short microtubules are able to move in a rapid and explicit way within the axons [14]. The microtubule severing increases its mobility and is important for the formation of the new axonal branches, and dendritic sprouting. Microtubule-severing activity is important for the development and maintenance of the nervous system [15]. Copy number variants including multi-exonic deletions/duplications in the *SPAST* gene are a frequent cause of autosomal dominant hereditary spastic paraplegia type 4. They are present in approximately 20% of patients suffering from HSP and comprise 30% of all alterations in the *SPAST* gene [16,17]. The increase in the number of CNVs detected in patients has sparked interest in the mechanism underlying the formation of large deletions and duplications. Mechanisms of genomic copy number change include (i) non-allelic homologous recombination (NAHR), (ii) non-homologous end joining (NHEJ), (iii) microhomology-mediated repair (MMR) mechanisms, such as fork-stalling and template switching (FoSTeS), microhomology-mediated end-joining (MMEJ), and microhomology-mediated break-induced replication (MMBIR) [18]. The formation of rearrangements is mediated by repetitive sequence: low copy repeats (LCRs) and interspersed repetitive sequences like long interspersed nuclear elements (LINEs) or *Alu* elements. Microhomology-mediation between *Alu* elements is implicated in the generation of wide-size CNVs in the genome or intermediate (100–1000 bp) in their size. *Alu-*specific microhomology is present in the NAHR recombination mechanism (*Alu*–*Alu* recombination) and is sufficient to provide a substrate for the microhomology-mediated processes of FoSTeS/MMBIR [3,5]. One hundred sixty-three *Alu* family members exist in the 112,252 bp region encompassing the *SPAST* gene and represent 36% of this sequence [3]. *Alu*-rich genomic architecture of *SPAST* predisposes to a variety of genome rearrangements and may explain the appreciable percentage of mutations causing SPG4. Up to date *Alu-*specific nonallelic microhomology recombination and complex rearrangements occurred by multiple *Alu*-facilitated template switches have been reported in *SPAST* [3,4,5]. Here we present five additional deletions and one duplication found in SPG4 patients, where the nonallelic microhomology recombination was the mechanism of the mutation. Together with previously reported cases, our results show the *Alu-*specific microhomology-mediated generation of intragenic CNV is the predominant mechanism generating rearrangements in *SPAST.* The *Alu* density at the *SPAST* gene locus is shown in Figure 2.

The presented research also indicates a loss of function mechanism in some, but not all, patients with *SPAST* gene deletions. We performed the quantitative real-time PCR analysis and three different fragments of the *SPAST* gene, spanning the exons 1–3, 6–8, and 12–15, respectively, were amplified. At least one of the primer sets was complementary to the part of the *SPAST* gene, where the deletion was located, so we knew that this level of transcription corresponded to only one allele of the gene, without deletion. The relative quantification was calculated between different amplicons in each patient. The obtained results show similar transcription levels in all three amplicons in patients with (i) deletion of 1–3 exons, (ii) deletion of 1–4 exons, (iii) deletion of 1–9 exons, and (iv) deletion of 10–12 exons. It suggests that the whole transcript, where the deletion was localized was degraded by the nonsense-mediated mRNA pathway. Hence the loss of function mechanism contributes to SPG4 due to the microrearrangements in the *SPAST* gene. The spectrum of mutations associated with SPG4 suggests that the loss of function mechanism and insufficient activity of the microtubule-severing lead to corticospinal axon degeneration. However, we assume that haploinsufficiency is not the only mechanism contributing to hereditary spastic paraplegia by microrearrangements in the *SPAST* gene. The results obtained in the presented study show a reduced transcriptomic level only in the part of the *SPAST* where the deletion occurs. This has been shown in patients with (i) deletions of 4–7 exons, (ii) deletions of 5–7 exons, (iii) exon 8 deletion, (iv) deletion of exons 8–17, (v) deletion of exons 10–17, and (vi) deletion of exons 14–16. We assume that, in these cases, an alternative splice site can be created, followed by transcription of the *SPAST* in which only part of the gene is missing. The *SPAST* open reading frame has two initiation codons. As a result, two isoforms of the *SPAST* gene may be transcribed. The M87 isoform protein contains 530 amino acids and is the predominant isoform in all tissues, whereas the longer M1 isoform (616 amino acids) is only detectable in the spinal cord [12,19]. However, Solowska et al. reported three dominant negative mutations in *SPAST*, which do not affect the enzymatic activity or expression level but result in the dominant negative activity of the M1 *SPAST* isoform [20]. The presented study indicates that microrearrangements in *SPAST* lead to hereditary spastic paraplegia not only via haploinsufficiency.

## Figures and Tables

**Figure 1 ijms-25-05008-f001:**
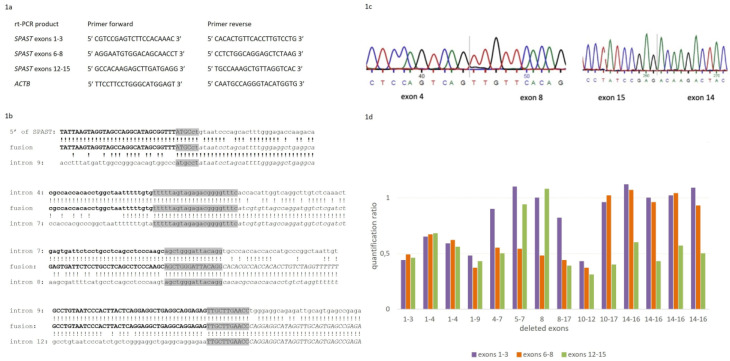
(**a**) Shows the sequence of primers used in real-time quantitative PCR; (**b**) fusion sequences identified in patients with deletions of exons 1–9, 5–7, 8, and 10–12 in the *SPAST* gene, which revealed nucleotide homology resulting in non-allelic homologous recombination; (**c**) cDNA sequences showing the absence of exons 5–7 in the *SPAST* gene and an altered sequence in which exon 14 is present after exon 15, resulting from tandem duplication in *SPAST*; (**d**) evaluation of the *SPAST* gene mRNA in patients with various exonic deletions in the gene. The quantitative ratio was measured for three amplification products—exons 1–3, exons 6–8, and exons 12–15 of the *SPAST* gene. The presented ratio was calculated as the ratio of mRNA levels between the studied individuals and the healthy controls.

**Figure 2 ijms-25-05008-f002:**
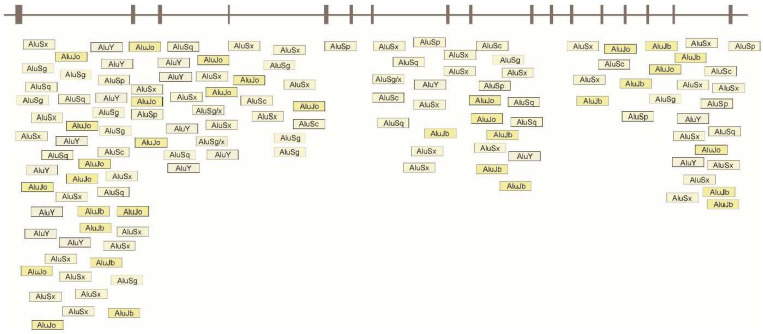
*Alu* sequence architecture in the *SPAST* gene locus.

**Table 1 ijms-25-05008-t001:** The table shows the clinical evaluation of patients from families with multi-exon deletions/duplication in the *SPAST* gene and summarizes the data collected from each family: breakpoint analysis, cDNA analysis, or gene expression analysis.

Family ID	*SPAST* Exonic Deletion/Duplication	Numer of Affected Family Members	Breakpoint Analysis	cDNA	Gene Expression Analysis	Age at Onset	Clinical Assesment	SPRS Score
Fam73	exon 1 deletion	2	-	-	-	45	na	na
Fam84	exon 1 deletion	1	-	-	-	20	pure spastic paraplegia	na
Fam137	exon 1 deletion	1	-	-	-	51	pure spastic paraplegia	12
Fam10	exons 1–3 deletion	2	-	-	NMD	27–35	pure spastic paraplegia	27
Fam40	exons 1–4 deletion	2	-	-	NMD	18–47	pure spastic parapegia	2–15
Fam100	exons 1–9 deletion	3	NAHR	-	NMD	8–40	pure spastic paraplegia	17–20
Fam23	exons 4–7 deletion	7	NAHR	breakpoint sequence	non NMD	1–44	spastic paraplegia, polyneuropathy in one patient	37–41
Fam38	exons 5–7 deletion	3	NAHR	breakpoint sequence	non NMD	12–52	spastic paraplegia, dysarthria in one patient	14–25
Fam17	exon 8 deletion	5	NAHR		non NMD	6–50	pure spastic paraplegia	18–31
Fam81	exons 8–17 deletion	3	-	-	non NMD	1–18	spastic paraplegia, myoclonic seizures in one patient	32
Fam87	exons 10–12 deletion	3	NAHR	breakpoint sequence	NMD	5–40	pure spastic paraplegia	25
Fam14	exons 10–17 deletion	2	-	-	-	5	pure spastic paraplegia	na
Fam39	exons 10–17 deletion	1	-	-	non NMD	4	pure spastic paraplegia	28
Fam37	exons 14–15 duplication	2	NAHR	breakpoint sequence	-	15	pure spastic paraplegia	14
Fam2	exons 14–16 deletion	6	-	breakpoint sequence	non NMD	10–35	pure spastic paraplegia	23–27
Fam36	exons 14–16 deletion	2	-	breakpoint sequence	non NMD	25–50	pure spastic paraplegia	46
Fam149	exons 14–16 deletion	5	-	-	non NMD	7–50	pure spastic paraplegia, head dropping in one patient	3–41

Abbreviations: NAHR—non-allelic homologous recombination; NMD—nonsense-mediated decay; SPRS—spastic paraplegia rating scale.

## Data Availability

The data that support the findings of this study are available on reasonable request from the corresponding author.

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
