# Peer review of "SPAST Intragenic CNVs Lead to Hereditary Spastic Paraplegia via a Haploinsufficiency Mechanism"

_ijms, 2024, doi:10.3390/ijms25095008_

Round 1
Reviewer 1 Report
Comments and Suggestions for Authors
The authors study intragenic CNVs in the SPAST gene in patients with Hereditary Spastic Paraplegia. The findings reported in this study are interesting and it adds to the previously reported CNVs in the SPAST gene but the presentation of the data should be improved. Following are the concerns,
1. The abstract of the article has many spelling mistakes and grammatical errors. For example in line 13, withing for within and presented for present.
2. The study was done with samples from 50 patients, the authors haven't included a comprehensive list of the genotype analysis for each patient.
3. In figure 1b, the table shows the different mutations detected in the patient samples but the corresponding sequencing data is missing in the paper. This information is essential.
4. In figure 1c, alignment of sequences in intron 4 and 7 is shown. A similar alignment of other regions encompassing the different deletions listed in the figure 1b must be showed.
5. In figure 1e, the authors show the quantification ratio of deleted exons but this data is confusing as there is no data on the axis.
Comments on the Quality of English LanguageEnglish language quality must be improved as there are a few grammatical errors and spelling mistakes.
Author Response
Thank you for your valuable comments. We have made adjustments according to your suggestions:
- Basic data on the clinical evaluation of patients with SPAST gene microrearrangements were included. Table 1 has been added to the manuscript.
- The abstract of the article has been improved and spelling mistakes have been corrected.
- A cDNA sequencing procedure has been added to the Materials and Methods section. The cDNA sequencing results are mentioned in the Results – Breakpoints and cDNA sequencing section.
- We added 3 more alignment analyzes of the breakpoints identified in SPG4 patients in Figure 1c. They show nucleotide homology between intronic sequences, which results in non-allelic homologous recombination – NAHR.
- The SPAST gene expression analysis section was revised. In figure 1d the y axis scale was added.
Reviewer 2 Report
Comments and Suggestions for Authors
In this manuscript authors cover an important genetic aspect of the CNV in SPAST gene. The paper is very interesting. I would suggest to add clinical information: are there any differences between patients with insertions or deletions? This wuld bring more clinical and pratical information that may help clinicians in the clinical setting and wouls broaeden the genotype-pehotype correlations
Author Response
Thank you for your valuable suggestions. According to report, we performed the following changes:
Basic data on the clinical evaluation of patients with SPAST gene microrearrangements were included. Table 1 has been added to the manuscript.
Reviewer 3 Report
Comments and Suggestions for Authors
In the present work from Elert-Dobkowska et al., in line with previously reported cases, the authors show that the Alu-specific microhomology-mediated generation of intragenic CNV is the predominant mechanism generating rearrangements in SPAST gene. They also show that haploinsufficiency is not the only mechanism contributing to hereditary spastic paraplegia by microrearrangements in the SPAST gene since, for some rearrangements, quantitative real time PCR revealed reduced transcriptomic level only in the part of the SPAST, where the deletion occurs. This finding would suggest that an alternative splice-site can be created, followed by transcription of the SPAST, in which only part of the gene is missing or an alternative open reading frame can be used.
The paper in well-written and the experiments are properly performed however it could be useful the employment of prediction tools to detect potential cryptic splice sites and transcript analysis in order to prove the hypothesis of alternative splicing isoforms. In addition a Westem Blot analysis on protein samples from patient’ cells expressing SPAST protein could be useful to prove the hypothesis of an alternative ORF.
Author Response
Thank you for your valuable comments. We are grateful for the comment concerning the prediction analysis of the cryptic splice sites in the SPAST gene. I would like to mention that this topic will be continued and we plan to use firstly in silico tools as you suggested, and next the long-range sequencing to characterize the splicing in patients with microrearrangements and patients with single nucleotide splice site variants in the SPAST gene. We are going to consider using the Western blot in our future study.
Round 2
Reviewer 1 Report
Comments and Suggestions for Authors
The authors have addressed the concerns raised in the first review sufficiently.
Comments on the Quality of English LanguageEnglish language quality can be improved further.
Author Response
Thank you for your comments.
We performed minor English revision as you suggested . Spelling mistakes and articles were corrected.
Reviewer 3 Report
Comments and Suggestions for Authors
The paper has been revised. A very useful table has been added to the text. However, Figure 1 should be further fixed (panel d should not be placed above panel c). I guess that the first paragraph of the discussion has been erroneously canceled.
Author Response
Thank you for your comments.
Accordingly to your suggestions Figure 1 has been edited. Particular elements were organized and accordingly described.
Furthermore, we checked that in the submitted version of the manuscript, any of the Discussion paragraphs is missed.